# On the Dynamics of Flexible Wings for Designing a Flapping-Wing UAV

**Renan Cavenaghi Silva** [1,*,†] and **Douglas D. Bueno** [2,†]

1    Department of Mechanical Engineering, São Paulo State University—UNESP, Avenida Brasil, 56-Centro, Ilha Solteira 15385-000, SP, Brazil
2    Department of Mathematics, São Paulo State University—UNESP, Avenida Brasil, 56-Centro, Ilha Solteira 15385-000, SP, Brazil; douglas.bueno@unesp.br
*    Correspondence: cavenaghi.silva@unesp.br; Tel.: +55-(018)-991-222-679
†    These authors contributed equally to this work.

**Abstract:** The increasing number of applications involving the use of UAVs has motivated the research for design considerations that increase the safety, endurance, range, and payload capability of these vehicles. In this article, the dynamics of a flexible flapping wing is investigated, focused on designing bio-inspired UAVs. A dynamic model of the Flapping-Wing UAV is proposed by using 2D beam elements defined in the absolute nodal coordinate formulation, and the flapping is imposed through constraint equations coupled to the equation of motion using Lagrange multipliers. The nodal coordinate trajectories are obtained by integrating the equation of motion using the Runge–Kutta algorithm. The imposed flapping is modulated using a proposed smooth function to reduce transient vibrations at the start of the motion. The results shows that wing flexibility yields significant differences compared to rigid-wing models, depending on the flapping frequency. Limited amplitude of oscillation is obtained when considering a non-resonant flapping strategy, whereas in resonance, the energy levels efficiently increase. The results also demonstrate the influence of different flapping strategies on the energy dissipation, which are relevant to increasing the time of flight. The proposed approach is an interesting alternative for designing flexible, bio-inspired, flapping-wing UAVs.

**Keywords:** FWUAV; dynamics; ANCF; flexible wing; vibrations; flapping





## 1. Introduction

The growing demand from individuals and companies for the use of unmanned aerial vehicles (UAVs) in civil and military applications is motivating research into design considerations that increase the safety, endurance, range, and payload capability of these vehicles. In contrast with conventional configurations (e.g., fixed-wing, helicopter, and multi-rotor), the bio-inspired Flapping-Wing Unmanned Aerial Vehicle (FWUAV) is inspired by bird flight and is a promising candidate for improving these requirements by mechanically flapping artificial wings to generate lift and thrust. Research on FWUAVs is motivated by the outstanding performance observed in birds, such as migrating species [1], which inspires understanding of the fundamentals of flapping flight to propose efficient solutions for the design of UAVs. Di Luca et al. [2] demonstrate that folding the wing reduces drag during cruise and, with asymmetric folding, it functions as an aileron. Pesavento and Wang [3] optimize the flapping kinematics of an insect wing model and discuss how flapping flight is 27% more efficient than a conventional steady wing by taking advantage of unsteady aerodynamic effects from the interaction of the wing with its own wake during wing reversal. In addition to that, an FWUAV configuration is safer for urban operation due to the lack of fast-rotating parts [4], which is a characteristic that suits this type of vehicle for applications that involve human–machine interaction.

The flapping motion in an FWUAV is usually achieved using mechanical links that connect the wing to a four-bar mechanism [5]. However, as Gerdes et al. [6] point out,

once the linkages are assembled, the input amplitude of the flapping is fixed, and only the frequency remains to be controlled [7]. On the other hand, birds adapt their flapping pattern depending on the flight phase. Cochran et al. [8] provide data that migratory species (*Catharus ustulatus*, *Catharus fuscescens*, and *Hylocichla mustelina*) use a higher amplitude and higher frequency continuous flapping during climbing and intermittent flapping consisting of periods of continuously flapping followed by periods of gliding during the cruise phase. Therefore, the design of an FWUAV capable of controlling both the flapping frequency and the amplitude is desired.

Typical FWUAV models usually assume a rigid wing [9], whereas flexibility can be used to design efficient vehicles. Flapping flight is a very complex phenomenon, and the rigid-wing model may not capture important aerodynamic features, such as the aeroelastic opening of the primaries during upstroke [10]. Heathcote et al. [11] show that a limited amount of flexibility also benefits the propulsive efficiency of the flapping wing. Therefore, modeling the flexibility of the wing during the flapping motion is an important step toward a better understanding of the fundamentals of flapping flight.

The advantage of modeling the wing while neglecting the flexibility is that the velocities along the wing span are obtained solely from the wing kinematics. However, as applications scale in range, payload, and other requirements, the need to design efficient flyers requires reducing the weight of the components, also including the structure. This implies selecting materials and optimizing their geometry so that the structure is light and, consequently, flexible. In addition to that, increasing the wing-span also has a significant effect on the structure's flexibility, which renders this type of analysis relevant for flapping wings with a high aspect ratio. On the other hand, if flexibility is not fully considered when designing an FW-based UAV, the resulting aerial vehicle is not an efficient flyer due to the effect of local vibrating modes.

Vanella et al. [12] present a model composed of two rigid links coupled with a lumped torsional spring representing the chord-wise flexibility. The flapping motion is prescribed by the position and pitch of the section, and an exponential smooth function is used to avoid transient vibrations. Yin and Luo [13] impose harmonic translation and rotation motion on a model of a flexible wing section and discuss the increase in the lift-to-drag ratio when considering its flexibility. Tian et al. [14] present the model for insect wings considering chord-wise deformation. The flapping input is applied by considering a harmonic function with a phase difference between the flapping and pitching. The model developed by each of these authors is mainly concerned with characterizing the influence of chord-wise flexibility. On the other hand, this article evaluates the effect of span-wise flexibility on the response of the structure during the flapping motion. The structural-based force magnitudes are significantly more influential compared to aerodynamic forces [15], and this is a typical strategy for FWUAV design [16], mainly focused on low airspeed flight. The results demonstrate that span-wise flexibility has a significant effect on the motion of each section depending on the flapping frequency and can be conveniently considered to design efficient flapping-wing-based UAVs. In this sense, this present article presents the effects of flexibility for various flapping parameters.

This paper presents an approach to demonstrate the influence of flexibility on the dynamics of a flapping wing. Note that nature has developed flying animals over many evolutionary years, and they exhibit efficient flight, mainly in terms of energy consumption, maneuver envelope, and static and dynamic stability. However, in terms of bio-inspired UAV design, the approaches in the literature involve flapping-wing-based formulations, and they usually consider rigid structures. On the other hand, flexibility is an important characteristic in the efficiency of birds, and it is inherent to the materials employed to manufacture aerial vehicles, mainly because they need to be light. To this end, a model of an FWUAV with flexible wings is proposed. The wing structure is discretized using absolute coordinates, which is an adequate formulation for computing the response of the structure subject to the large expected flapping amplitude of motion. The equation of motion of the wing performing a flapping motion is obtained with a multi-body system approach, where

the motion constraints are embedded into the equations through Lagrange multipliers. The results demonstrate how the flexibility can be considered for designing an efficient flapping wing focused on bio-inspired UAVs. The remainder of this article is organized as follows. Section 2 presents the nomenclature and parameters of the proposed flexible flapping-wing model. Section 3 presents the modeling methodology and the equation of motion of the FWUAV using the absolute nodal coordinate formulation (ANCF). Section 4 presents the results of different flapping strategies applied to the model. Section 5 presents the final conclusions and suggestions for future research.

## 2. Methodology

The dynamic model consists of a flapping joint at $O$, which defines the origin of the wing coordinate system. The wing is modeled by two portions: an inner and rigid wing of length $L_i$ attached to the flapping joint, and a flexible outer wing with non-deformed length $L_o$ attached to the other end of the rigid wing. The rigid wing attached to the flapping joint defines a time-dependent flapping angle $\theta(t)$ with respect to the $x$-direction. This angle is directly controlled, and it prescribes the motion of the wing by considering different flapping strategies. A typical bird and the bio-inspired mechanical model of the FWUAV wing are presented in Figure 1a,b, respectively, where the points of interest of the model are associated with their counterparts on the bird wing.

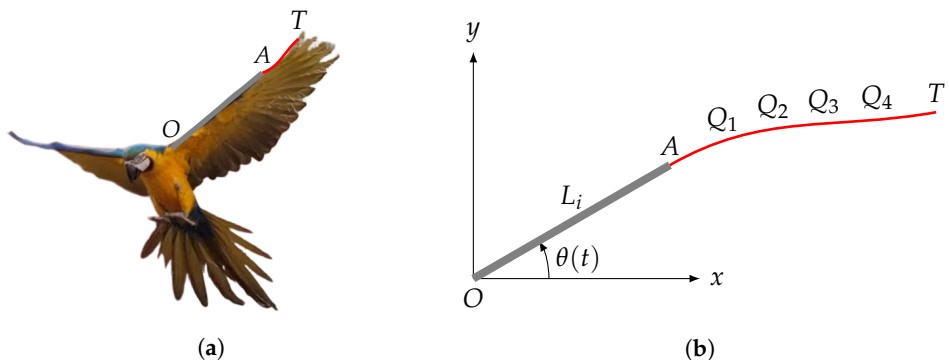

|  |  |
|:---:|:---:|
| (**a**) | (**b**) |

**Figure 1.** (**a**) Picture of a common bird in Brazil (*Ara ararauna*) and the model representing the wing structure overlapped. (**b**) Flexible, bio-inspired, flapping-wing model. The points $A$, $Q_1$, $Q_2$, $Q_3$, $Q_4$, and $T$ define points of interest in the flexible wing.

Different points along the flexible wing are labeled; the rigid-flexible interface by $A$, the tip of the wing by $T$, and intermediate points located at 20%, 40%, 60%, and 80% of the flexible wing length as $Q_1$, $Q_2$, $Q_3$, and $Q_4$, respectively. The positions of these last ones are chosen arbitrarily.

*Rigid Wing Motion*

The rigid wing flapping pattern is obtained from the wing kinematics. The position of an arbitrary point in the rigid wing is computed by:

$$\mathbf{r}_{RW}(t) = \begin{Bmatrix} x_{RW}(t) \\ y_{RW}(t) \end{Bmatrix} = \begin{Bmatrix} l \cos \theta(t) \\ l \sin \theta(t) \end{Bmatrix} \tag{1}$$

where $0 \le l \le L_i$ defines the position of the interest point along the wingspan. The velocity is obtained by differentiating Equation (1) with respect to time:

$$\dot{\mathbf{r}}_{RW}(t) = \begin{Bmatrix} \dot{x}_{RW}(t) \\ \dot{y}_{RW}(t) \end{Bmatrix} = \begin{Bmatrix} -l\dot{\theta}(t) \sin \theta(t) \\ l\dot{\theta}(t) \cos \theta(t) \end{Bmatrix}. \tag{2}$$

## 3. Flexible Wing Dynamics

The Absolute Nodal Coordinate Formulation (ANCF) is used to obtain the equation of motion from the elastic wing. This method describes the dynamics of structures and

multibody systems using absolute coordinates and slopes as nodal coordinates [17]. Since infinitesimal rotations are not used, the equations of motion are valid for the analysis of structures subjected to large deflections and deformations [18]. Furthermore, the mass matrix is constant, while the Coriolis and centrifugal forces are identically equal to zero [19]. On the other hand, a non-linear expression for the elastic forces is obtained. Yoo et al. [20] obtained accurate trajectories of a thin cantilevered beam with an attached rigid body under free vibrations obtained using the ANCF and a high-speed image acquisition system. The results were extended by considering vertical vibrations forced through the base motion [21].

The position of an arbitrary particle from the beam is represented using the ANCF, such as

$$\boldsymbol{r}(t) = \boldsymbol{S}(\xi)\boldsymbol{e}(t),\tag{3}$$

where $\boldsymbol{S}(\xi)$ and $\boldsymbol{e}(t)$ are the shape function and the nodal coordinates, respectively. The shape function of the 2D beam element using the ANCF is defined by:

$$\boldsymbol{S}(\xi) = \begin{bmatrix} s_1\boldsymbol{I} & s_2\boldsymbol{I} & s_3\boldsymbol{I} & s_4\boldsymbol{I} \end{bmatrix},\tag{4}$$

with $\xi$ denoting the normalized length of the element, i.e., $\xi = x/l$, where $l$ is the length of the element. $\boldsymbol{I}$ is an identity matrix with 2 rows and 2 columns, and $s_1 = 1 - 3\xi^2 + 2\xi^3$, $s_2 = l(\xi - 2\xi^2 + \xi^3)$, $s_3 = 3\xi^2 - 2\xi^3$, and $s_4 = l(\xi^3 - \xi^2)$ are the polynomial interpolation coefficients of the shape function [22].

The nodal coordinates of the two-dimensional ANCF beam element are the position of its end points with respect to the absolute coordinate system and the slope from the beam center line with respect to the coordinate system. Denoting by the subscripts $(\cdot)_A$ and $(\cdot)_B$ the end nodes of the element, the nodal coordinate vector of the 2D beam element is defined by:

$$\boldsymbol{e}(t) = \left\{ \boldsymbol{r}_A(t)^\top \quad \frac{\partial \boldsymbol{r}_A(t)}{\partial \boldsymbol{x}}^\top \quad \boldsymbol{r}_B(t)^\top \quad \frac{\partial \boldsymbol{r}_B(t)}{\partial \boldsymbol{x}}^\top \right\}^\top,\tag{5}$$

and the velocity and acceleration of an arbitrary particle in the element are obtained by, respectively,

$$\dot{\boldsymbol{r}}(t) = \boldsymbol{S}\dot{\boldsymbol{e}}(t)\tag{6}$$

$$\ddot{\boldsymbol{r}}(t) = \boldsymbol{S}\ddot{\boldsymbol{e}}(t),\tag{7}$$

which allows one to compute the kinetic energy of the element by considering the following equation:

$$\mathcal{T} = \frac{1}{2}\dot{\boldsymbol{e}}(t)^\top \left[ \int_V \rho \boldsymbol{S}^\top \boldsymbol{S} dV \right] \dot{\boldsymbol{e}}(t),\tag{8}$$

where $\rho$ is the specific mass of the material, $V = Al$ is the volume of the element, and $A$ is the cross section area of the element. The integral in Equation (8) results in the constant mass matrix $\boldsymbol{M}$, which is reproduced below for convenience [17]:

$$\boldsymbol{M} = m \begin{bmatrix} \frac{13}{35} & 0 & \frac{11l}{210} & 0 & \frac{9}{70} & 0 & -\frac{13l}{420} & 0 \\ & \frac{13}{35} & 0 & \frac{11l}{210} & 0 & \frac{9}{70} & 0 & -\frac{13l}{420} \\ & & \frac{l^2}{105} & 0 & \frac{13l}{420} & 0 & -\frac{l^2}{140} & 0 \\ & & & \frac{l^2}{105} & 0 & \frac{13l}{420} & 0 & -\frac{l^2}{140} \\ & & & & \frac{13}{35} & 0 & -\frac{11l}{210} & 0 \\ & \text{symm.} & & & & \frac{13}{35} & 0 & -\frac{11l}{210} \\ & & & & & & \frac{l^2}{105} & 0 \\ & & & & & & & \frac{l^2}{105} \end{bmatrix} \tag{9}$$

where $m = \rho V$ is the mass of the element. The potential energy resulting from the flexibility of the wing is given by [22]

$$\mathscr{U} = \mathscr{U}_l + \mathscr{U}_t = \frac{1}{2} \int_0^l EA\epsilon^2 ds + \frac{1}{2} \int_0^l EI\kappa^2 ds, \tag{10}$$

where $\mathscr{U}_l$ and $\mathscr{U}_t$ are the potential energies due to the longitudinal and flexural deformations, respectively. This term is cumbersome to compute since both the longitudinal deformation $\epsilon$ and the curvature $\kappa$ are functions of the nodal coordinates, leading to highly nonlinear terms for the elastic forces. Different assumptions for these parameters result in different models for the elastic forces. If the longitudinal deformation can be assumed constant along the element, models L1 and L3 may be employed [23].

The mechanical energy of the system is computed by:

$$\mathscr{E} = \mathscr{T} + \mathscr{U}. \tag{11}$$

Note that the longitudinal stiffness matrix is a function of the nodal coordinates, since it depends on the value of the assumed constant longitudinal deformation across the element $\bar{\epsilon}_l = \frac{\sqrt{(e_5 - e_1)^2 + (e_6 - e_2)^2}}{l} - 1$, with $e_i$ representing the components from $\boldsymbol{e}$. The elastic force due to the longitudinal strain is

$$\boldsymbol{Q}_l = \boldsymbol{K}_l \boldsymbol{e}, \tag{12}$$

such that

$$\boldsymbol{K}_l(\boldsymbol{e}) = \frac{EA}{l} \bar{\epsilon}_l \begin{bmatrix} \frac{6}{5} & 0 & \frac{l}{10} & 0 & \frac{-6}{5} & 0 & \frac{l}{10} & 0 \\ & \frac{6}{5} & 0 & \frac{l}{10} & 0 & \frac{-6}{5} & 0 & \frac{l}{10} \\ & & \frac{2l^2}{15} & 0 & \frac{-l}{10} & 0 & \frac{-l^2}{30} & 0 \\ & & & \frac{2l^2}{15} & 0 & \frac{-l}{10} & 0 & \frac{-l^2}{30} \\ & & & & \frac{6}{5} & 0 & \frac{-l}{10} & 0 \\ & \text{symm.} & & & & \frac{6}{5} & 0 & \frac{-l}{10} \\ & & & & & & \frac{2l^2}{15} & 0 \\ & & & & & & & \frac{2l^2}{15} \end{bmatrix}. \tag{13}$$

The elastic force due to transverse deformation is

$$Q_t = K_t e, \tag{14}$$

such that

$$K_t = \frac{EI}{l^3} \begin{bmatrix} 12 & 0 & 6l & 0 & -12 & 0 & 6l & 0 \\ & 12 & 0 & 6l & 0 & -12 & 0 & 6l \\ & & 4l^2 & 0 & -6l & 0 & 2l^2 & 0 \\ & & & 4l^2 & 0 & -6l & 0 & 2l^2 \\ & & & & 12 & 0 & -6l & 0 \\ & & \text{symm.} & & & 12 & 0 & -6l \\ & & & & & & 4l^2 & 0 \\ & & & & & & & 4l^2 \end{bmatrix}. \tag{15}$$

Then, the elastic force term is represented by the following force vector:

$$f = -(K_l(e) + K_t)e. \tag{16}$$

### 3.1. Flapping Motion Constraints

The flapping motion is imposed using constraint equations that are written in terms of the time $t$ and the nodal coordinates $e(t)$. The constraint consists of prescribing the position and slope at the joint $A$ from Figure 1 through the following constraint vector $\mathbf{\Phi}$:

$$\mathbf{\Phi}(t, e) = \left\{ \begin{array}{c} e_A - d_A(t) \\ \dfrac{de_A}{dx} - \theta_A(t) \end{array} \right\}, \tag{17}$$

where $d_A(t)$ and $\theta_A(t)$ are the imposed position and slope of the joint, respectively, and where the subscript $(\cdot)_A$ indicates point A of the model. Equation (17) represents a set of rheonomic constraints that must be satisfied for all instants of time [24]. Differentiating Equation (17) two times with respect to time, the following equation is obtained:

$$\ddot{\mathbf{\Phi}}(t, e, \dot{e}) = \left\{ \begin{array}{c} \ddot{e}_{A_x} + L_i\ddot{\theta}(t) \sin\,\theta(t) + L_i\dot{\theta}^2(t) \cos\,\theta(t) \\ \ddot{e}_{A_y} - L_i\ddot{\theta}(t) \cos\,\theta(t) + L_i\dot{\theta}^2(t) \sin\,\theta(t) \\ \dfrac{\partial \ddot{e}_A}{\partial x} + \ddot{\theta}(t) \sin\,\theta(t) + \dot{\theta}^2(t) \cos\theta(t) \\ \dfrac{\partial \ddot{e}_A}{\partial y} - \ddot{\theta}(t) \cos\,\theta(t) + \dot{\theta}^2(t) \sin\,\theta(t) \end{array} \right\}, \tag{18}$$

from which the following expression is obtained:

$$A\ddot{e}(t) = b(t), \tag{19}$$

by considering the the vector $b(t)$ and the matrix $A$ given by, respectively:

$$b(t) = \left\{ \begin{array}{c} -L_i\ddot{\theta}(t) \sin\,\theta(t) - L_i\dot{\theta}^2(t) \cos\,\theta(t) \\ L_i\ddot{\theta}(t) \cos\,\theta(t) - L_i\dot{\theta}^2(t) \sin\,\theta(t) \\ -\ddot{\theta}(t) \sin\,\theta(t) - \dot{\theta}^2(t) \cos\theta(t) \\ +\ddot{\theta}(t) \cos\,\theta(t) - \dot{\theta}^2(t) \sin\,\theta(t) \end{array} \right\}, \tag{20}$$

$$A = \begin{bmatrix} I_{4\times4} & 0_{4\times4} \end{bmatrix}, \tag{21}$$

where $I$ and $0$ indicate the identity and zero matrices, respectively. The subscript indicates the dimension of the matrix.

### 3.2. Equation of Motion

The dynamic model of the constrained mechanical system representing the flexible flapping wing is represented by the following system of differential and algebraic equations (DAE):

$$M\ddot{e} + A^\top \Lambda = f \tag{22}$$

$$\Phi(t, e) = 0, \tag{23}$$

where $\Lambda$ is the Lagrange multiplier vector and $f$ is the vector containing the elastic forces. However, the solution is not easily obtained from the DAE system, and a convenient approach consists of reducing the index of the problem [25] by differentiating the constraint equation to obtain an equivalent model in the form of ordinary differential equations (ODE) [26,27]; see Equation (19).

$$M\ddot{e} + A^\top \Lambda = f \tag{24}$$

$$A\ddot{e} = b. \tag{25}$$

The solution to Equation (25) is also a solution to the original system, and it is obtained with a standard algorithm for solving the ODE system, such as the Runge–Kutta algorithm [28]. However, the resulting Equation (23) is mildly numerically unstable, and its solution may require the use of constraint-stabilizing parameters if the trajectories are solved for a long period of time [26]. In this case, the following equivalent constraint equations are considered instead [29]:

$$A\ddot{e} + 2\alpha_{bg}\dot{\Phi} + \beta_{bg}^2 \Phi = b, \tag{26}$$

where $\alpha_{bg} > 0$ and $\beta_{bg}$ are the two stabilizing parameters that modify the constraint-associated hypersurface into an attractor such that the solution trajectory, that is, $e(t)$, approximately satisfies the constraint equations. The following system represents the dynamic of the FWUAV subjected to the flapping motion constraints:

$$\ddot{e} = M^{-1}\left(f - A^\top \Lambda\right), \tag{27}$$

$$AM^{-1}\left(f - A^\top \Lambda\right) = b - 2\alpha_{bg}\dot{\Phi} - \beta_{bg}^2 \Phi \tag{28}$$

$$\Lambda = \left(AM^{-1}A^\top\right)^{-1}\left(-b + 2\alpha_{bg}\dot{\Phi} + \beta_{bg}^2 \Phi + M^{-1}f\right). \tag{29}$$

Since the term $\left(AM^{-1}A^\top\right)$ is constant over time, it is computed only once during the simulation. Then, the acceleration of the system is obtained by substituting the Lagrange multiplier back into Equation (27), resulting in the following expression:

$$\ddot{e} = M^{-1}\left(f - A^\top\left(AM^{-1}A^\top\right)^{-1}\left(-b + 2\alpha_{bg}\dot{\Phi} + \beta_{bg}^2 \Phi + M^{-1}f\right)\right). \tag{30}$$

Note that the solution to Equation (30) requires us to define a set of initial conditions compatible with the system constraints, i.e., $e(0)$, such that both $\Phi(0, e(0)) = 0$ and $\dot{\Phi}(0, e(0)) = 0$. This ensures that the system configuration starts on the constraint hypersurface.

The advantage of the presented FWUAV model is its versatility. In addition to the advantages associated with the ANCF, including the accuracy of analyzing the system response to large amplitude of motions, the possibility of directly including the constraint equations allows for different configurations of FWUAVs to be rapidly modeled, speeding up the development process. Note that the accuracy of ANFC-based approaches have been presented by different authors in the literature, such as Yoo et al. [20], Yoo et al. [21], Farokhi et al. [30], and others.

Note that the equation of motion neglects the aerodynamic forces. This is a typical strategy for small vehicles, mainly when a low-speed flight is evaluated. Otherwise, aerodynamic effects combined with structural dynamics can generate static and dynamic phenomena, such as aeroelastic divergence and flutter. However, at low speeds in a cruise phase of flight, the structural forces are significantly more influential than aerodynamic forces are [16].

### 3.3. Modal Parameters

The elastic forces are linearized with respect to the equilibrium configuration to obtain the modal parameters of the flexible wing modeled using the ANCF. Then, the modal parameters from the wing considering a clamped-free boundary condition are obtained from the generalized eigenvalue problem:

$$\det\left(\frac{\partial f}{\partial e}\Big|_{e=\tilde{e}} - \omega_n^2 M\right) = 0 \tag{31}$$

with $\tilde{e}$ denoting the equilibrium configuration from the elastic wing.

### 3.4. An Efficient Way to Start the Flapping Motion

An efficient approach to starting the flapping motion is to consider an incremental amplitude from zero to the desired amplitude of motion. This behavior can be achieved by using a modulation function (MF). This strategy is relevant to reduce the influence of the higher-order vibration modes in relation to the first bending mode on the wing response due to the control input. In practice, in an abrupt motion, i.e., performing the first cycle of motion considering the desired flapping amplitude, higher-order vibration modes are sufficiently energized to contribute to the motion. On the other hand, in performing the first cycles while considering an amplitude that increases until achieving the flapping amplitude, the energy from the input is more concentrated in the frequency range of the first bending mode, and the flapping motion is more efficient in terms of energy usage. In this sense, the higher-order vibration modes represent undesired vibrations, in terms of efficiency of flapping.

There are different strategies for defining an input of increasing amplitude. In particular, if an exponential MF is used (cf. [12]), an asymptotic convergence to the flapping amplitude can be verified. However, in terms of a practical implementation, this leads to a computational burden for an embedded system with limited resources, since a modulation value and its time derivative need to be computed for all instants during the flight. On the other hand, an alternative approach is introduced in this article based on a proposed modulating function that converges to the flapping amplitude in a finite interval. The proposed MF is parameter-dependent, and the values can be conveniently defined to obtain different strategies to start the flapping motion.

The modulation strategy in this paper consists of defining a smooth transition between rest and motion, which is denoted by Smooth Harmonic Flapping (SHF). Modulation improves the quality of the flapping by reducing the excitation of higher frequencies and undesired vibrations. The smooth function $s(t)$ is defined as:

$$s\left(\frac{t-a}{b-a}\right) = \begin{cases} 0, & \text{if} \quad t \le a \\ 3\left(\frac{t-a}{b-a}\right)^2 - 2\left(\frac{t-a}{b-a}\right)^3, & \text{if} \quad a < t < b, \\ 1, & \text{otherwise} \end{cases} \tag{32}$$

where $a$ and $b$ are constant parameters that modulate the starting and end points of the transition. Figure 2 illustrates the envelope defined by Equation (32).

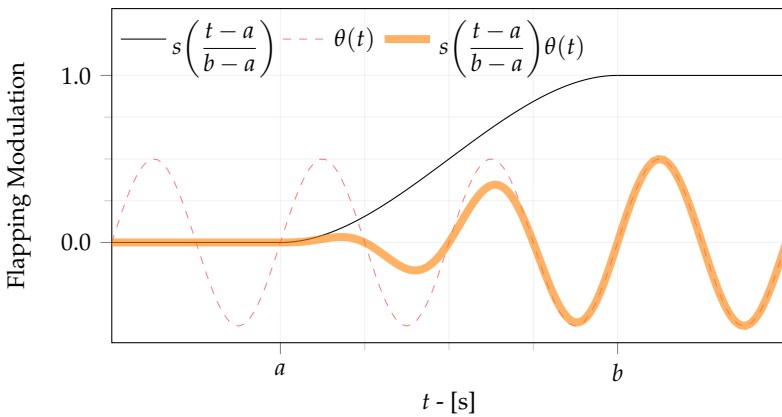

**Figure 2.** Illustration of the proposed smooth function.

The particular harmonic flapping (HF) strategy, i.e., $\theta(t) = a_F \sin (\omega_F t)$, is obtained if the modulating parameters chosen are $a = -1$ and $b = 0$. Then, the flapping angle, angular velocity, and angular acceleration with respect to the time are defined by, respectively:

$$\theta(t) = s(t)a_F \sin (\omega_F t), \tag{33}$$

$$\dot{\theta}(t) = \dot{s}(t)a_F \sin(\omega_F t) + s(t)a_F \omega_F \cos (\omega_F t), \tag{34}$$

$$\ddot{\theta}(t) = \ddot{s}(t)a_F \sin(\omega_F t) + \dot{s}(t)a_F \omega_F \cos (\omega_F t) - s(t)a_F \omega_F^2 \sin (\omega_F t). \tag{35}$$

The inverted smooth function $\tilde{s}(t) = 1 - s(t)$ is used for the transition between flapping and non-flapping. Combining the $s(t)$ and $\tilde{s}(t)$ functions, a smooth step is obtained, used to define the intermittent Resonant Flapping (RF) strategy for Model R, i.e., $\theta(t) = s(t)\tilde{s}(t)a_F \sin (\omega_F t)$.

The position and velocity in the $y-$ direction are normalized with respect to the flapping parameters by considering the following equations:

$$\tilde{y} = \frac{y}{(L_i + L_o) \sin (a_F)}, \tag{36}$$

$$\dot{\tilde{y}} = \frac{\dot{y}}{(L_i + L_o)\omega_F a_F \cos (a_F)}, \tag{37}$$

where the denominator in Equation (36) corresponds to the vertical position of the tip of an equivalent rigid wing.

## 4. Results and Discussion

The proposed approach is demonstrated for a flapping-wing model considering the physical and geometric parameters shown in Table 1. The Young modulus $E$ and the material density $\rho$ are considered to match the properties of an aluminum 7075-T6 alloy, which is a material widely used in the aerospace industry.

Although flying feathers usually have a varying geometry along their axis [31], which changes the modal properties of the structure, it is assumed that the cross section in the model has a constant shape of area $A$ and inertia $I$ throughout the span to keep the analysis concise. However, the ANCF can also be applied to model tapered and twisted structures such as wind turbines [32].

**Table 1.** Flapping-wing model parameters.

| Parameter | Value | Unit |
|-----------|-------|------|
| $L_i$ | 0.55 | m |
| $L_o$ | 1.25 | m |
| $A$ | $9.6 \times 10^{-5}$ | $m^2$ |
| $I$ | $5.12 \times 10^{-10}$ | $m^4$ |
| $\rho$ | 2810 | $kg/m^3$ |
| $E$ | $71.7 \times 10^9$ | $N/m^2$ |

The first three vibration modes are considered, and their corresponding frequencies are obtained by solving the generalized eigenvalue problem, Equation (31), which correspond to $f_1 = 5.222$ Hz, $f_2 = 32.728$ Hz, and $f_3 = 91.650$ Hz, respectively. They numerically correspond to the analytical values found in Table 2 of [33], p. 103. The flapping frequency is normalized with respect to the frequency of the first mode for convenience, such that $\tilde{f}_F = \frac{f_F}{f_1}$, where $f_F = 2\pi\omega_F$ is the flapping frequency converted to hertz. The normalized time is defined as $\tilde{t} = \tilde{f}_F t$.

Table 2 presents a summary for different flapping-wing models. The sixth-order Runge–Kutta (RK6) algorithm is used to integrate the equation of motion [34]. The second column of the table indicates the flapping strategy. The third and fourth columns indicate the smooth function modulating parameters. Columns 5 and 6 present the flapping amplitude and normalized flapping frequency, respectively. The parameters to discretize the elastic wing using the ANCF and the time step for the RK6 algorithm to integrate Equation (30) are presented in columns 7 and 8, respectively. The HF, SHF, and R flapping strategies are defined in the following.

**Table 2.** Summary of the parameters for each flapping-wing model.

| Model | Strategy | $a$ | $b$ | $a_F$ | $\tilde{f}_F$ | $n_e$ | $\Delta t$ |
|-------|----------|-----|-----|-------|---------------|-------|------------|
| 0 | HF | $-1$ | 0 | $25°$ | 0.2 | 5 | $10^{-6}$ |
| 1 | SHF | 0 | 1 | $25°$ | 0.2 | 5 | $10^{-6}$ |
| 2 | SHF | 0 | 2 | $25°$ | 0.6 | 5 | $10^{-6}$ |
| R | RF [1] | $[0, 0.4]$ | $[1, 1.4]$ | $5°$ | 1 | 5 | $10^{-6}$ |
| 3 | SHF | 0 | 3 | $15°$ | 1.8 | 12 | $10^{-6}$ |

[1] The first and second values of the intervals refer to $a$ and $b$ for the smooth and inverted functions, respectively.

Model 0 consists of the harmonic flapping (HF) strategy. The system is initially at rest, and the harmonic flapping input defined in Equation (33) is applied, considering $a = -1$ and $b = 0$. The normalized vertical position and velocity of each point of interest in Model 0 are presented in terms of the normalized time $\tilde{t}$ in Figure 3a,b, respectively. A variation of one unit from the normalized time corresponds to one flapping cycle. The trajectory from point $A$, represented by the red line, is prescribed from the flapping kinematics; thus, it represents the input applied to the model. On the other hand, the points $Q_i$, $i = 1, \ldots, 4$, and $T$ are obtained by solving the equations of motion.

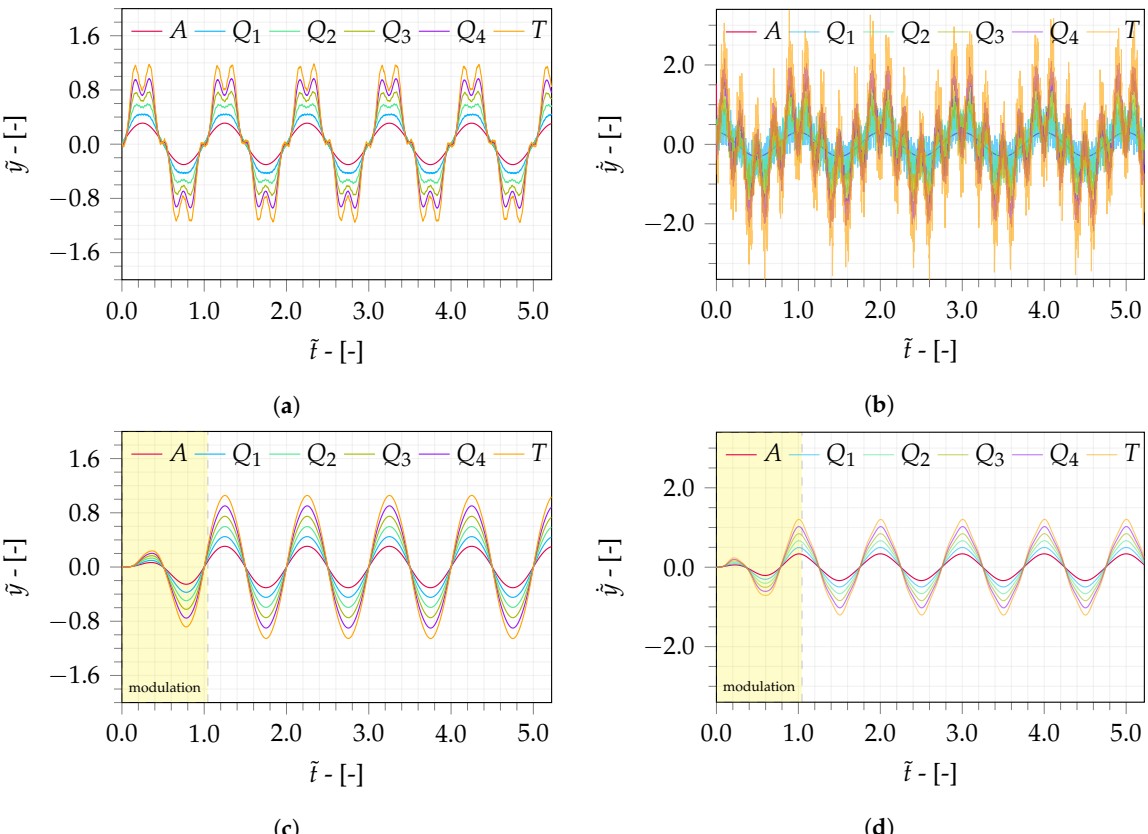

**Figure 3.** Trajectories of the flapping wing considering the HF and SHF flapping strategies. The normalized vertical position (**a**) and velocity (**b**) of Model 0. The normalized vertical position (**c**) and velocity (**d**) of Model 1.

Model 1 (see Table 2) is identical to Model 0 except that the input is modulated. Variations in the normalized vertical position and normalized velocity from points of interest with respect to normalized time $\tilde{t} = f_F t$ are presented in Figure 3c,d, respectively. Note that Model 1 does not show the higher frequency oscillations of Model 0 shown in Figure 3a due to the input modulation. The period in which the flapping is modulated by $s(t)$ is represented in Figures 3c,d in the highlighted region. Thus, if the wing is initially at rest, such as during gliding, a smooth transition is required to avoid the vibrations shown by the trajectories of Model 0.

Alternatively, the trajectories of the wing are presented in the phase portrait as illustrated in Figure 4a,c,e. Each orbit illustrates the trajectory of the points of interest after the modulation period. The orbits outlined using dashed lines correspond to the position and velocity of an equivalent point in a rigid wing, computed through kinematics (Equations (1) and (2)). The points illustrated in the phase portrait are intersections of the trajectories with Poincaré sections defined at multiples of the flapping period. The points in these sections represent the mode shape of the wing during the flapping motion.

Figure 4b,d,f illustrate the trajectory of the points of interest in the *xy* plane. The deflection of the elastic wing at two snapshots that occurs when the tip has its maximum and minimum deflections is presented by the red curve. These snapshots contrast with the configuration from an equivalent rigid wing computed through the wing kinematics shown by the straight gray line.

The solution for Model 2 is obtained considering the duration of the smooth flapping as $(b - a) = 2$ s, as only one second was insufficient to reduce the influence of the transition. The trajectories in the phase portrait are presented in Figure 4c, and the corresponding flapping pattern is shown in Figure 4d.

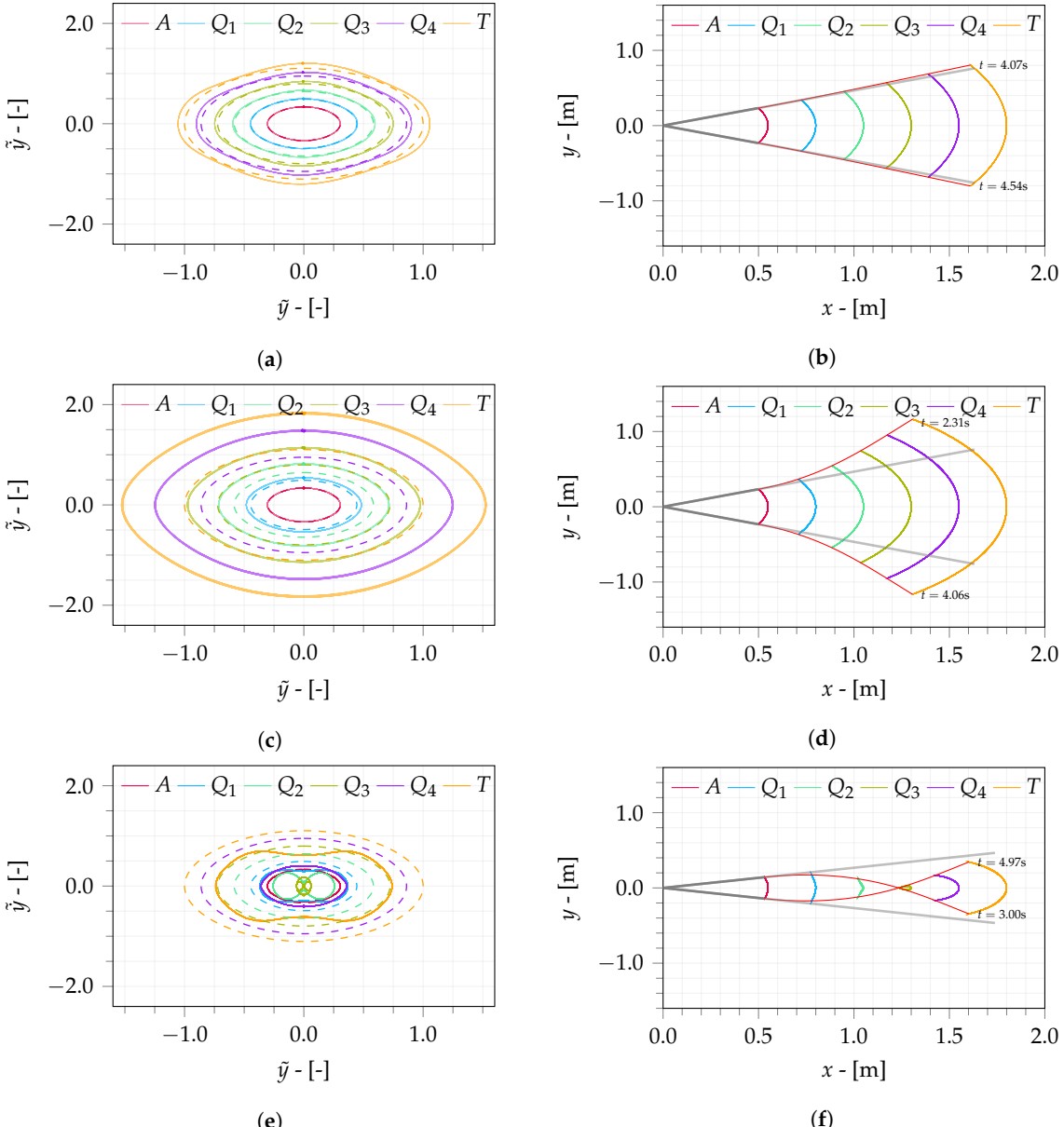

**Figure 4.** Trajectories of the system for various flapping parameters. (**a**) Phase portrait and (**b**) flapping pattern in the $xy$ plane of Model 1. (**c**) Phase portrait and (**d**) flapping pattern in the $xy$ plane of Model 2. (**e**) Phase portrait and (**f**) flapping pattern in the $xy$ plane of Model 3.

The position and velocity from Models 1 and 2 are increased in comparison with the rigid wing, which is illustrated by the larger orbits from all points of the elastic wing when compared with the orbits of the corresponding points in the rigid wing operating at the same flapping conditions. In practice, the effect of flexibility is to increase both the amplitude and velocity of oscillation of points in the flexible wing in comparison with the rigid-wing dynamics.

When comparing the pattern in the $xy$ plane between Models 1 and 2, the increase in the flapping frequency yields additional bending, significantly increasing the amplitude of oscillation for the points of interest. Specifically, for the snapshots in Figure 4b,d, the maximum vertical position of the tip increases from 0.804 m in Model 1 to 1.163 m in Model 2, a relative increase of 44.65%. In contrast to the maximum displacement of the rigid wing, 0.760 m, the increases in Models 1 and 2 are 5.69% and 52.88%, respectively.

Model 3 corresponds to the dynamics for a flapping frequency above the first natural frequency. The trajectories depicted in Figure 4e demonstrate that the outer points of

interest $Q_3$, $Q_4$, and $T$ have a phase difference with respect to the inner points $A$, $Q_1$, and $Q_2$, such that, while the inner sections are in the upstroke phase, the outer sections are partially executing the downstroke phase. In addition, the orbits of these points are reduced in comparison with the point of an equivalent rigid wing. Figure 4f illustrates two snapshots of the wing deflection.

The RF strategy (Table 2) consists of actuating the flapping joint in resonance with the first bending mode until sufficient energy is introduced into the system. Then, the flapping input stops, and the motion from the elastic wing persists due to its free dynamics. Figure 5a shows the phase portrait of the wing during resonant flapping. The orbit of $A$ collapses to a point in the phase portrait due to the stopping of the flapping input. Unlike Model 3, all points perform at the same upstroke or downstroke phase. However, the positions where they intersect the Poincaré sections are slightly different from the computed resonant frequency. The flapping pattern in the $xy$ plane is presented in Figure 5b.

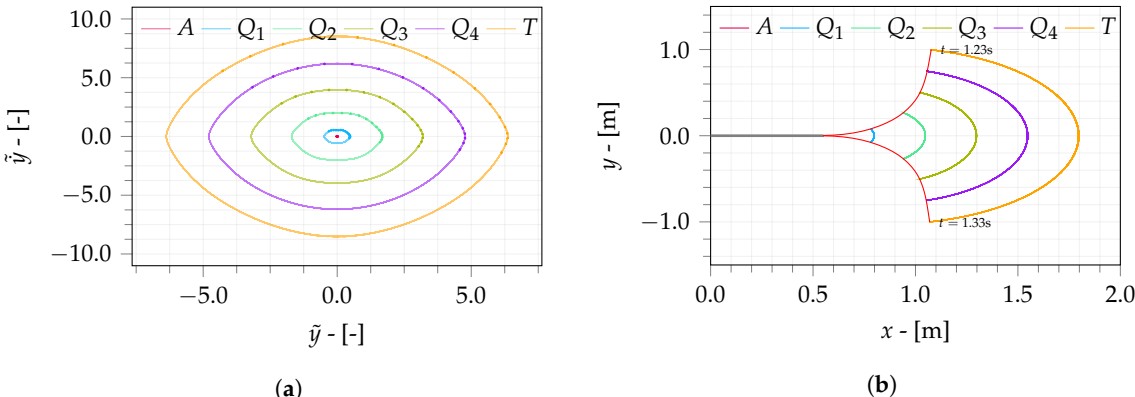

(a)　　　　　　　　　　　　　　　　　　　　　　　　(b)

**Figure 5.** Model R trajectories in the (**a**) phase portrait and (**b**) $xy$ plane.

During non-resonant flapping, the mechanical energy of the system is limited and continuously oscillates, as shown in Figure 6a, while the flapping input is applied. This result shows the inefficiency of this flapping strategy, since the energy transmitted through the imposed motion is used to reduce the mechanical energy of the system and is not likely to be converted back to battery capacity. Thus, for a fixed flapping frequency, the only possibility to increase the mechanical energy of the wing is to increase the amplitude of the flapping input.

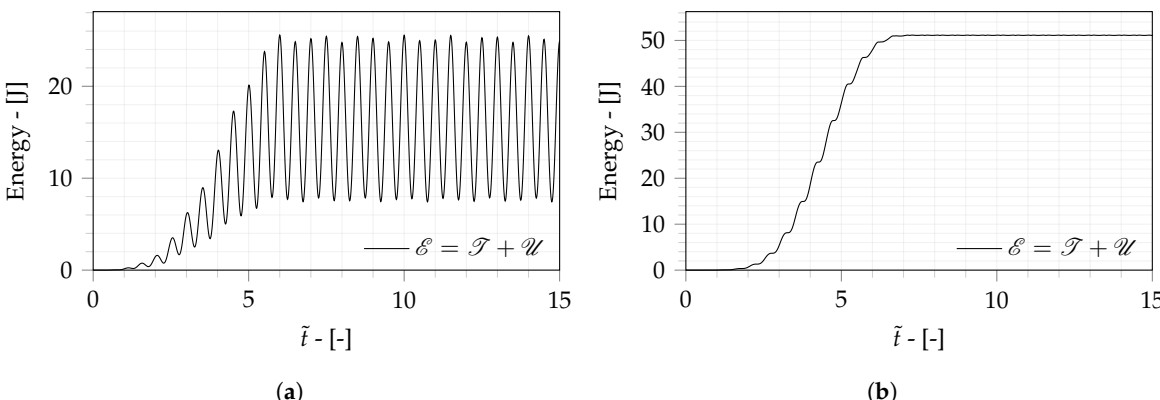

(a)　　　　　　　　　　　　　　　　　　　　　　　　(b)

**Figure 6.** Mechanical energy of (**a**) Model 2 and (**b**) Model R.

Unlike the non-resonant flapping strategy, the mechanical energy of the wing of Model R continuously increases while in resonance, as shown in Figure 6b. Then, after the flapping stops, the mechanical energy is conserved, since there is no dissipation included in this bio-inspired wing model. This distinguishing feature allows for a low-amplitude input flapping result at arbitrary amplitudes of oscillations in the flexible wings. This benefits,

for instance, the inclusion of solar panels in the rigid sections of the wing to generate electric energy and increase the flight time, since the low flapping amplitude allows them to operate under better conditions of light incidence.

The resonance-based strategy, Model R, consists of using intermittent flapping with a frequency equal to the first bending mode of the flexible wing. The results indicate that, even with a small flapping input amplitude, it is possible to continuously increase the energy stored in the flexible wing to obtain large amplitude oscillations. Then, when sufficient mechanical energy is stored, the flapping input is stopped, and the flexible wing remains flapping through its free dynamic. Intermittent resonant flapping is a promising strategy for increasing the endurance of the FWUAV, since energy is efficiently converted to mechanical energy in the form of the desired flapping motion. However, there are different mechanisms of energy dissipation in a physical mechanical system, as well as in biological species [35] (e.g., viscoelastic damping, aerodynamic friction, and induced drag), such that, after some period of time, the deflection amplitude of the flexible wing decreases.

Analysis of the effect of damping requires modeling mechanisms of energy dissipation, including viscoelastic damping in the ANCF model of Equation (30) as in [36], using complementary information obtained using system identification techniques to estimate the damping ratios [20]. Furthermore, in the presence of energy dissipation, the flapping amplitude in resonance is also limited by the damping ratio. Therefore, in the presence of energy dissipation, resonant flapping is applied until a desired amplitude of oscillation is obtained; then, the amplitude from the flapping input motion is reduced to provide the energy loss from dissipation during each flapping cycle. This flapping strategy encourages the implementation of a driving mechanism where both the amplitude and frequency from the flapping can be controlled, such as by using a servo-motor for accurate control of the position and speed. This is an interesting topic for further investigations using the approach of designing flapping-wing-based UAVs proposed herein.

The results of the proposed approach consider a flapping motion assuming deflection in the *xy* plane. In addition to that, avians also rotate their entire wing in the shoulder joint and the outermost feathers through the wrist joint in a combination that tends to be optimal. Försching and Hennings [10] discuss that the biological wing has its elastic center located ahead of the aerodynamic center to assist the actuated joints in rotating the wing, thus reducing the input torque required to drive the wing. Inclusion of the torsional degree of freedom requires us to consider a wing model in a three-dimensional space, which increases the modeling complexity.

Note that this proposed modeling includes only the bending motion. On the other hand, the inclusion of torsional dynamics can change the efficiency of the flapping wing at non-resonant frequencies, whereas it does not affect the efficiency at the resonant frequency once the modes are uncoupled from each other. At non-resonant frequencies, a combined bending/torsion flapping response is observed, instead of a pure bending motion. The stiffer mode (i.e., bending or torsion) introduces a small contribution to the resulting motion in relation to the more flexible mode because the focus of this proposed approach is to consider the frequency range until the first mode. Then, if the torsional mode corresponds to a higher resonant frequency in relation to the bending mode, the influence of the torsion is small.

## 5. Conclusions

This work presents the modeling of an FWUAV with a flexible wing using the ANCF. The constraint equations required to define the flapping motion of a bio-inspired FWUAV are obtained. A technique of index reduction is used to obtain the equation of motion as a system of ODEs. The trajectories of the different models are evaluated by integrating the equation of motion using the Runge–Kutta algorithm. A comparison of the trajectories from Models 0 and 1 reveals that modulating the flapping input motion with a smooth function is an effective way to reduce transient vibrations caused by the flexible wing. In addition to that, combining the modulation parameters conveniently defines different

flapping strategies, i.e., HF, SHF, and RF. Then, the flapping frequency is varied to analyze the effects of flexibility on the flapping pattern. Considering the wing flexibility, Models 1 and 2 show an increase in the amplitude of deflection of all points of interest in comparison with the trajectories of these points in an equivalent rigid wing. In particular, the tip deflection increases by 5.69% and 52.88% when flapping at 20% and 60% of the first natural frequency, respectively. Increasing the flapping frequency beyond the first natural frequency, i.e., Model 3, yields reduced orbits in comparison with the rigid-wing model. In addition to that, the flexible wing vibrates similarly to the second mode shape, with part of the wing in the upstroke and the remaining performing the downstroke. Flapping in resonance results in significant deflections, and it requires intermittent flapping to limit the oscillation amplitude after the structure is sufficiently energized. This article has shown that limited mechanical energy is stored by the wing with a non-resonant flapping strategy. Thus, the only parameter remaining to obtain a larger oscillation from the elastic wing is to increase the amplitude of the imposed flapping angle, which depends on the mechanical construction of the FWUAV, so this is not always possible. On the other hand, the intermittent flapping strategy can be applied in resonance until the desired amplitude of motion is obtained. Then, the flapping input is interrupted, and the flexible wing remains flapping due to its free dynamics.

**Author Contributions:** Conceptualization, R.C.S. and D.D.B.; Funding acquisition, D.D.B.; Investigation, R.C.S.; Methodology, R.C.S.; Project administration, D.D.B.; Software, R.C.S.; Supervision, D.D.B.; Visualization, R.C.S.; Writing—original draft, R.C.S.; Writing—review & editing, D.D.B. All authors have read and agreed to the published version of the manuscript.

**Funding:** The authors appreciate the support under grant numbers: 2023/04325-3 and 2022/03128-7 from the São Paulo Research Foundation (FAPESP). The second author thanks the National Council for Scientific and Technological Development (CNPq), grant number 314151/2021-4. This study was financed in part by the Coordenação de Aperfeiçoamento de Pessoal de Nível Superior-Brasil (CAPES)-Finance Code 001.

**Data Availability Statement:** The data presented in this study are available on request from the corresponding author. The data are not publicly available due to privacy.

**Conflicts of Interest:** The authors declare no conflict of interest.

## Abbreviations

The following abbreviations are used in this manuscript:

| | |
|---|---|
| ANCF | Absolute Nodal Coordinate Formulation |
| DAE | Differential Algebraic Equation |
| FWUAV | Flapping Wing Unmanned Aerial Vehicle |
| HF | Harmonic Flapping |
| MF | Modulation Function |
| ODE | Ordinary Differential Equation |
| RF | Resonant Flapping |
| SHF | Smooth Harmonic Flapping |

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
