# Peer review of "On the Dynamics of Flexible Wings for Designing a Flapping-Wing UAV"

_drones, doi:10.3390/drones8020056_

Round 1

Reviewer 1 Report

Comments and Suggestions for Authors

This paper presents a method for efficiently computing the flapping motion of a flexible wing, which is a potentially efficient propulsion method for UAVs. The authors then use the model to analyse different flapping strategies: high amplitude, non-resonant and low amplitude, resonant. The resonant strategie seems to provide advantages from the point lf view of energy efficiency, and the main reasons are also discussed.

The reasoning seems sound and the equations of motion are well derived. The control strategies are well selected and the discussion and main conclusions are correct and relevant. However, some of the simplifications may limit the validity of the results and the practical applications of the method. I agree with neglecting the aerodynamic forces when computing the flapping motion in some configutations (much lower Reynolds number than the one presented here), but I am not completely sure if it is sensible to do so in this study. Could you please provide more insight into that? Also, as far as I understand, the model is derived for a constant section, uniform moment of inertia wing. Could you please comment on the implications of having a wing with a span-varying section? Finally, torsion is also neglected in this study, although I am not sure that it is realistic to do so with a thin and flexible wing, whith a (probably) very low torsional stiffness and potentially extreme rotation angles near its tip if its aspect ratio is not only big, but also moderate. Can you explain the limitations of the model in that aspect and how do you expect to mitigate them?

I have no other issues with the document: it is also clear and well presented.

Reviewer 2 Report

Comments and Suggestions for Authors

This article is well written and appropriate figures are given. The stated primary purpose of the paper is to study the energy efficiency of different flexible, flapping wing strategies. I think the authors have done a good job in showing this by comparing several strategies. My only concern is the inability of being able to assess the accuracy or limitations of this research. This is because, although aspects of flapping strategies are discussed with previous peer-reviewed research publications, the end results are not compared or discussed. I believe the paper would be greatly improved if this could be done, at least at some level. It is incomplete without this.

Reviewer 3 Report

Comments and Suggestions for Authors

This paper mainly introduces the flexible modeling process of fluttering wings, simulates and analyzes the effects of several fluttering strategies on fluttering wings. However, the scientific significance of the research content is insufficient, and is difficult to reflect the innovation of the research. I recommend this manuscript must be major revised for the following reasons:

1. The language application needs to be strengthened. There are many statement problems in the text, including the title of the subject is not clear.

2. The article’s content level is unclear, and needs to re-examine the planning: the abstract has many discussions about the background of the research; the introduction section is interspersed with a lot of later research content; the meaning of second section is not clear; In section 4, there are many simulation processes that should be explained in advance, the results and discussions are scattered after the explanation of each simulation stage, which makes this section very lengthy and hard to understand; The summary in the conclusion section is not enough and there are a lot of discussion parts in this section, which should be in the ‘results and discussion’ section.

3. The subject of the research is not clear: the title is the study of the flexible dynamics for designing the flapping wing, but the abstract mentions the study of the effects of different control strategies on the flapping wing, while the conclusion mentions that it provides a method for the design of the flapping wing.

4. The main work of this paper is the flexible modeling process, and simulation analysis of several fluttering strategies on fluttering wings. Although the article is rich in content, but the whole text does not summarize the conclusions. The content of this article is difficult to reflect the research value of the paper.

Comments on the Quality of English Language

Minor editing of English language required

Round 2

Reviewer 1 Report

Comments and Suggestions for Authors

The authors have properly answered all my questions and comments and I think that the manuscript is now worth publishing.